# Understanding the genetic sex-determining mechanism in *Hyla eximia* treefrog inferred from H-Y antigen

**Aidet Ruiz[1], Guadalupe Cárdenas[2], Desiderio Velasco[2], Luis Ramos**[1]*

**1** Department of Reproductive Biology, Instituto Nacional de Ciencias Médicas y Nutrición Salvador Zubirán, México City, Mexico, **2** Genética y Estudios Cromosómicos y Moleculares S.C., México City, Mexico

* luis.ramost@incmnsz.mx

## Abstract

Genetic sex-determining mechanisms have been extensively elucidated in mammals; however, the sex chromosomes, sex-determining genes, and gene regulatory networks involved in sex differentiation remain poorly understood in amphibians. In this study, we investigated the sex-determining mechanism in the *Hyla eximia* treefrog based on karyotypic analysis and identification of H-Y antigen, a sex-linked peptide that is present in the gonads of the heterogametic sex (XY or ZW) in all vertebrates. Results show a diploid chromosome number 2n = 24 with homomorphic sex chromosomes. The heterogametic sex, ZW-female, were hypothesized based on H-Y antigen mRNA expression in female gonads (24,ZZ/24,ZW). The treefrog H-Y peptide exhibited a high percentage of identity with other vertebrate sequences uploaded to GenBank database. To obtain gene expression profiles, we also obtained the coding sequence of the housekeeping *Actb* gene. High H-Y antigen expression levels were further confirmed in ovaries using real-time polymerase chain reaction (RT-PCR) during non-breeding season, we noted a decrease in the expression of the H-Y antigen during breeding season. This study provides evidence that sex hormones might suppress H-Y antigen expression in the gonads of heterogametic females 24,ZW during the breeding season. These findings suggest that H-Y gene expression is a well-suited model for studying heterogametic sex by comparing the male and female gonads.

**Data Availability Statement:** All relevant data are within the manuscript and its Supporting information files.

**Funding:** The author(s) received no specific funding for this work.

## Introduction

In vertebrates, sex-determining mechanisms can be mainly subdivided into two types according to the nature of the main sex-determining factor [1, 2]: (1) genetic sex determination (GSD) in which the genomic factor plays a relevant biological role (such as chromosomal) or (2) environmental sex determination (ESD) in which the main factor is an environmental cue (such as temperature or light). In mammals (including humans) and birds, sex-determining mechanisms are controlled by XX/XY (XX females/XY male heterogamety) or ZZ/ZW (ZW females heterogamety/ZZ males) chromosomes, respectively [3–5]; which contain either a dominant gene on the heteromorphic sex chromosome (such as *SRY* in humans) or a dosage-

**Competing interests:** The authors have declared that no competing interests exist.

sensitive gene on the homomorphic sex chromosome (such as *DMRT1* in birds). In many reptiles and fish, the environment can determine or influence the sex of developing embryos [6]. Previous studies have demonstrated that the genetic influence of sex chromosomes (ZZ/ZW) can be overridden by high incubation temperatures, which causes ZZ male to female sex reversal in a dragon lizard [7]. In addition, multiple sex chromosome systems (X1X2Y and Z1Z2W) have been reported in mammals [8], reptiles (with both oviparous and viviparous clades) [9], and amphibians [10]. These data highlight the variability in chromosomic and genetic sex-determining mechanisms in vertebrate groups.

Unlike in some fish and reptiles displaying ESD, in amphibians, temperature and/or light are not relevant for sex determination because amphibian sex determination is genetically controlled [11–13]. Ruiz-Garcia et al commented that there is a lack of understanding [14] about the sex chromosomes, sex-determining genes, and the gene regulatory networks involved in sexual differentiation in the amphibia class. In this regard, amphibians (as with many fishes and reptiles) rarely exhibit strongly heteromorphic sex chromosomes and homomorphic sex chromosomes have been described in 95% of the species with a known karyotype [11, 15]. Both male (XX/XY) and female (ZZ/ZW) heterogamety can be found in amphibian species, although male heterogamety is more frequent [16].

An immunological approach for the identification of the heterogametic sex is often deduced from cell membrane-associated minor histocompatibility Y-specific (H-Y) antigen or cross-reactive antigen, which is present in the gonads of the heterogametic sex (XY or ZW) in all vertebrates that have been studied so far, including amphibians [17, 18]. Sex-linked H-Y antigen is a short peptide (11 amino acids) encoded by a segment of a gene called the "select mouse cDNA on Y" or *Smcy*/*SMCY* (also known as *HY*, *HYA*, *JARID1D*, or *KDM5D*), which was originally found on the Y chromosome of male mammals [19, 20] from mice and humans and which can lead to graft rejection of male donor cells in a female recipient [21, 22]. In non-mammalian vertebrates, the H-Y antigen is encoded by a fragment of an orthologous gene defined as *Smcx*/*SMCX* (also known as *MRXJ*, *MRX13*, *MRXSJ*, *XE169*, *MRXSCJ*, *JARID1C*, *DXS1272E*, or *KDM5C*; https://www.ncbi.nlm.nih.gov/gene/8284/ortholog/similargenes/?scope=7776). In eutherian mammals, both *SMCY*/*SMCX* genes diverged at least 120 million years ago and are highly homologous to each other [23, 24]. The results have permitted researchers to draw the conclusion that the H-Y antigen is a fragment of a gene that is extremely conserved during vertebrate evolution into fish, amphibians, reptiles, birds, and mammals [25]. Immunological H-Y antigen tests have been conducted in species, such as *Rana pipiens*, *Xenopus laevis* [26], *Bufo bufo*, *Pyxicephalus adspersus*, and *Triturus vulgaris* [25]; however, molecular assays in amphibia class and its gene regulation by sex hormones have not been done.

After focusing on the previously mentioned data and obtaining novel information on sex determination in the *Hyla eximia* treefrog, we applied real-time polymerase chain (qPCR) and hypothesized the genetic sex-determining mechanism in the *H. eximia* treefrog inferred from H-Y antigen based on a karyotype analysis in the adult stage, Sanger sequencing, and tissue expression during the non-breeding and breeding seasons. Furthermore, we isolated and sequenced the coding region of the *Actb* gene.

## Materials and methods

### Amphibian tissues and ethical statements

We collected 16 adult specimens (eight females and eight males) of the *H. eximia* treefrog (Anura: Hylidae, Fig 1A) in breeding season (June to August) during summer rainy and non-breeding season (November to April) from the Transverse Neovolcanic Ridge in central

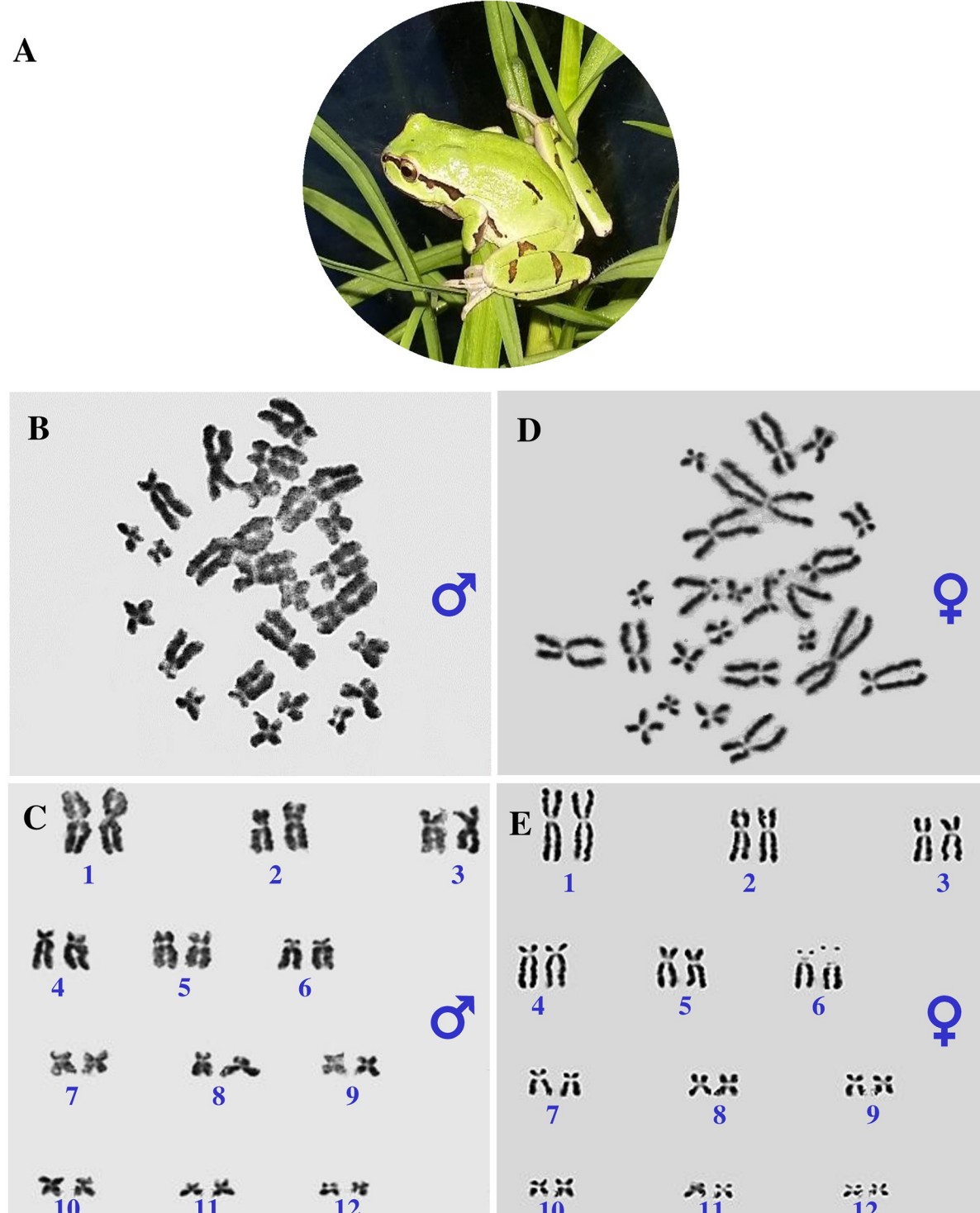

**Fig 1. Karyotyping, chromosome structure, and diploid metaphase chromosome sets of *Hyla eximia* treefrog. A** Adult specimen of the *H. eximia* treefrog (Anura: Hylidae). Male karyotype is shown in **B** and **C**, while the female karyotype is shown in **D** and **E**. The number of chromosomes identified in *H. eximia* was 24 in both sexes. Chromosome pairs were numbered from 1 to 12 based on the morphometric similarities. Karyotype stained with Giemsa showing the absence of heteromorphic sex chromosomes (**C** and **E**). The same chromosome number were observed between specimens of *H. eximia* of both sexes; therefore, the karyotype indicated a diploid number 2n = 24 for females and 2n = 24 for males. The identification of heterogametic sex was not visualized through karyotyping.

Mexico (19˚03'42.3"N 99˚14'57.2"W). Initially, all treefrogs were sexed according to size differences after which they were sacrificed by decapitation. Consequently, the phenotypic sex of the *H. eximia* treefrog was confirmed and determined based on the macroscopic identification of male and female gonads under a dissecting microscope (Bristoline, Feasterville, PA, USA). In this study, we only used the gonads of males and females in the non-breeding season to determine H-Y antigen expression and subsequently identify the impact of the sex hormones. Different tissues (testis, ovary, lung, brain, vocal sac, anterior and posterior muscles) were obtained, and the total RNA from each tissue was isolated. In this study, the gonads were used to analyze the H-Y antigen expression and determine the heterogametic sex. The other tissues were obtained for comparative purposes of gene expression. Samples were frozen immediately in liquid nitrogen and stored at −70˚C until further processing. All animals were collected under authorization from Secretaría de Medio Ambiente y Recursos Naturales (SEMARNAT, SGPA/DGVS/008330/18) and the Comisión Nacional de Áreas Naturales Protegidas (CON-ANP, F00.6.DRCEN/0866/18) approved the study and use of test samples from the *H. eximia* treefrog. The Animal Research Committee of our Institution approved the study for the care and use of animals (INCMNSZ, BRE-1940-19-25-1).

## Cytogenetic analysis

By cardiac puncture, karyotyping was performed using 30 μL of lymphocytes collected in sodium heparin. The cells were seeded in a culture medium containing Roswell Park Memorial Institute (RPMI) 1640 medium with 10% fetal calf serum, phytohemagglutinin, and 1% antibiotic-antimycotic (Penicillin–Streptomycin–Amphotericin). The mixture was incubated at 28˚C for 72 h. Twenty-five microliters of 0.04% PBS KaryoMAX™ Colcemid™ (Gibco, Life Technologies, NY, USA) were added to each cell culture after which the cultures were incubated at 30˚C for 2 h. The samples were centrifuged at 1000× g for 10 min. The supernatant was removed, 8 mL of 0.075 M KCl was added to each cell pack, and then the mixtures were incubated for 30 min at 30˚C. The mixtures were centrifuged at 1000× g for 5 min to obtain a cell pellet to which 8 ml of ethanol:acetic acid (3:1) solution was added after which the pellet was kept for 30 min at room temperature and centrifuged at 1000× g for 5 min. The cell pellet was resuspended, and 300 μL of ethanol:acetic acid solution was dropped onto a slide and allowed to dry. The chromosomes were stained with Giemsa (1:10) for 20 min. The chromosome analysis was performed using 15 metaphases for each sex (n = 15 for males and n = 15 for females) in the 100X immersion objective. Karyotyping was analyzed by the automated cell imaging system Leica Biosystems, using Cytovision DM2500. The chromosomes were analyzed with ImageJ software and classified based on the centromeric index (CI) according to Levan et al. [27].

## Isolation and expression of H-Y antigen from gonads

We isolated total RNA from the testes and ovaries of adult *H. eximia* using TRIzol reagent according to the manufacturer's protocol (Invitrogen, Carlsbad, CA, USA). The purity (260/280 = 1.8–1.9) and concentration (2 μg/μL) of each total RNA sample were determined spectrophotometrically (Beckman DU 650, Fullerton, CA, USA) based on the A260/A280 absorbance ratio. The integrity of total RNA was assessed from the 28S and 18S ribosomal RNA bands using 5 μg of total RNA on a 1.5% formaldehyde–agarose gel and visualized with ultraviolet (UV) light (Molecular Imager ChemiDoc XRS with Image Lab Software, BioRad, Hercules, CA, USA). cDNA synthesis was conducted using 2.0 μg/μL of total RNA using the instructions of the Maxima First Strand cDNA Synthesis Kit for reverse transcriptase (RT)-qPCR (Thermo Scientific, Vilnius, Lithuania). We designed a pair of primers (forward: 5′-

tgcgtcaaggaaagtggttat-3′; reverse: 5′-taggaagttgtactggaatgttctc-3′, amplicon size of 276 base pairs [bp]) to amplify the H-Y antigen sequence using PCR. The software PrimerQuest Tool (Integrated DNA Technologies–IDT, Iowa, USA; https://www.idtdna.com/pages) was employed to design specific primers for the H-Y antigen. RT-PCR was carried out in a final volume of 20 μL, which contained 4 μL of 5X Colorless GoTaq Flexi Buffer, 0.5 μL of dNTP mix (10 mM each), 1.0 μL of each primer (20 μM), 2 mM of MgCl2, 1.0 μL of dimethyl sulfoxide (DMSO), 0.5 U of GoTaq DNA polymerase (Promega, Madison, WI, USA), and 2 μL of cDNA. The RT-PCR amplification parameters (Veriti 96 well Thermal Cycler, Applied Biosystems, Austin, TX, USA) consisted of several steps: (1) one cycle at 94˚C for 3 min; 25 cycles at 94˚C for 30 s, 62˚C for 30 s, and 72˚C for 30 s; and 1 final extension at 72˚C for 3 min. The fragments were analyzed by 1.0% agarose gels electrophoresis and purified by Amicon Ultra-4 Centrifugal Filter Devices (Merck Millipore, Cork, Ireland). The RT-PCR products were sequenced in both directions as described previously [28] and then a multiple sequence alignment by ClustalW (https://www.genome.jp/tools-bin/clustalw) were carried out using several sequences submitted to Genbank (https://www.ncbi.nlm.nih.gov/gene/). Sanger sequencing data analysis were performed on an ABI-PRISM 310 genetic analyzer (Applied Biosystems, Foster City, CA, USA).

## Cloning and sequencing of Actb for qPCR

To conduct quantitative qPCR studies, we used the nucleotide sequence of β-actin (Actb). Based on the predictive sequence from *Rana temporaria* (XM_040357114.1) and *Bufo gargarizans* (XM_044302802.1) Actb, we synthesized four pairs of primers (forward: 5′-ctggctttgctggagatga-3′, reverse: 5′-tacgtccggaggcataca-3′; forward: 5′-atggctacagctgcttcttc-3′, and reverse: 5′- gatccacatctgctggaaagt-3′) and obtained a specific sequence from *H. eximia* by PCR. The PCR product was purified and sequenced as previously described [29]. The resulting nucleotide sequence was used to synthesize two gene-specific primers (reverse: 5′-catagctgtccttctgtcccattcc-3′ and forward: 5′-ggtaccaccatgtaccctggcattg-3) and amplify the untranslated regions (UTR) of the Actb cDNA using the Rapid Amplification of cDNA Ends (RACE) kit (Clontech, Mountain View, CA, USA) according to the manufacturer's instructions. Two pairs of primers were synthesized to amplify the total Actb cDNA. The PCR product was cloned and sequenced according to previous protocols [29].

## Expression profiles of H-Y antigen

Six tissue samples (gonads, lung, brain, vocal sac, anterior and posterior muscles) from males and females during the non-breeding and breeding seasons were evaluated to detect the gene expression of the H-Y antigen in *H. eximia*. For treefrog qPCR, total RNA was isolated as described previously. Reverse transcription was performed with the Maxima First Strand cDNA Synthesis Kit for RT-qPCR using 2.0 μg/μL of total RNA in a 20 μL reaction volume. RT-qPCR reactions were performed using the LightCycler TaqMan Master kit (Roche Diagnostics, Indianapolis, IN, USA) and 20 μM of each primer (forward: 5′-cgttgttacaagaagctgatgaagt-3′ and reverse: 5′-gttgtactggaatgttctctgcttc-3′), 10 μM of universal probe for H-Y antigen (probe #88 04689135001; Germany Roche Diagnostics, Mannheim, Germany), 4 μl of TaqMan Master LightCycler reaction mix, 5 μl of cDNA in a 20 μL final reaction volume. *Actb* (β-actin) was used as the internal reference gene (forward: 5′-aggagattgccgcacttgtt-3′ and reverse: 5′-atcatctccagcaaagccgg-3′; probe #21 04686942001; Germany Roche Diagnostics, Mannheim, Germany). PCR amplification was performed under specific conditions: (1) 95˚C for 10 min (2) 40 cycles of 95˚C for 10 s, (3) 60˚C for 30 s, (4) 72˚C for 1 s, and (5) a final cooling cycle of 40˚C for 10 s. A fluorescence reading was obtained at the end of each 72˚C step. qPCR

reactions were performed using the LightCycler 2.0 instrument (Roche Diagnostics, Indianapolis, IN, USA). mRNA expression levels were analyzed using the quantitative cycle threshold (Ct) value of the sample, the relative quantification method provided by the LightCycler software (version 4.05), and 2−ΔΔCt. Four independent biological replicates were used for each tissue sample.

### Statistical analysis

The expression profiles were carried out in four independent biological replicates. Descriptive statistics were assessed using means and standard deviations (SD) for each experiment. For the data analysis, statistical comparisons were tested by applying one-way analysis of variance (ANOVA) tests. The significance level was set at $p < 0.05$.

## Results

### Karyotyping and chromosome arrangement

Using cytogenetic approaches and sex-specific molecular markers, we characterized the sex determination mechanism (24,ZZ/24,ZW) in a treefrog. Twenty-four chromosomes were visualized in the females and males (S1 and S2 Figs in S1 File). In this study, we describe the morphometry of each chromosome pair from *H. eximia* (Fig 1B–1D), which is a diploid number, 2n = 24, representing 12 homomorphic pairs in male and female karyotype. All chromosomes contained two arms and showed a decrease in size, either metacentric, submetacentric, or acrocentric (Fig 1B and 1D). Chromosome morphology was metacentric for pairs 1 (CI = 47.3), 2 (CI = 40.5), 3 (CI = 42.7), 7 (CI = 46.9), 9 (CI = 42.2), 10 (CI = 47.1) 11 (CI = 42.9), and 12 (CI = 40.7); submetacentric for pairs 5 (CI = 38.6) and 8 (CI = 34.0); and acrocentric for pairs 4 (CI = 24.8) and 6 (CI = 25.2). We did not identify sex chromosomes. No heteromorphic sex chromosomes (morphologically differentiated) were identified based on either size or shape in the two sexes of *H. eximia* (Fig 1C and 1E).

### Sequence and characterization of treefrog H-Y antigen and Actb

Isolation, Sanger sequencing, and RT-PCR amplifications of specific sequences of the H-Y antigen from the ovaries (Fig 2) indicated that the female of this species was heterogametic (ZW). The treefrog H-Y antigen cDNA (Fig 2A) transcript exhibited an open reading frame comprising 33 base pairs (bp) and encoding a protein of 11 amino acids (Fig 2B). The theoretical molecular weight (MW) of the H-Y antigen protein was 1133.28 Da, and its isoelectronic point (pI) was 4.53. Based on comparative analyses and sequence alignments of H-Y antigen protein sequences, the deduced amino acid sequence of the H-Y antigen showed a highly conserved structure of 11 amino acids. The average amino acid identity values obtained from the *H. eximia* H-Y antigen exhibited 54.54% between vertebrates (from mammalia, aves, and reptilia) and 90.90% closely related species, such as amphibia class (Fig 2C). RT-PCR was performed in two different tissues from the *H. eximia* treefrog. As shown in Fig 2D, the mRNA expression of H-Y antigen was identified in ovaries (Ova), while in the male gonads (Test), such expression was absent from non-breeding season samples. This implies females represent the heterogametic sex. qPCR assays were performed to obtain greater precision and specificity concerning H-Y antigen mRNA expression. To quantify the relative mRNA expression levels of H-Y; the cDNA sequence encoding Actb was isolated, cloned, and sequenced from ovaries of the *H. eximia* treefrog. The full-length sequence of the Actb cDNA (GenBank accession number PP107949) was 1415 bp long, including a poly-A tail (Fig 3). Its 5′- and 3′-UTRs were 91 bp and 167 bp long, respectively. A putative polyadenylation signal (AATAAA) was found

**A**

Nucleotide sequences of the treefrog H-Y antigen

TGCGTCAAGGAAAGTGGTTATCGCGGGTACAGGCCGCCCTTGTCGCCGGACGTTCGGGTACACTCCAACAGATGCG
TTCGTTGTTACAAGAAGCTGATGAAGTGGCGGAGAGCCCTGCGGTGGAGAAGGCTTGTAGTGAACTGCAGGAACT
AATCTCCATTGCTCTCCGCTGGGAAGAAAAGGCTCAGATGTGTTTAGAAGCAAGGCAGAAGCATCCTCCTGCTACC
CTTGAGGCAATCATAAAGGAAGCAGAGAACATTCCAGTACAACTTCCTA

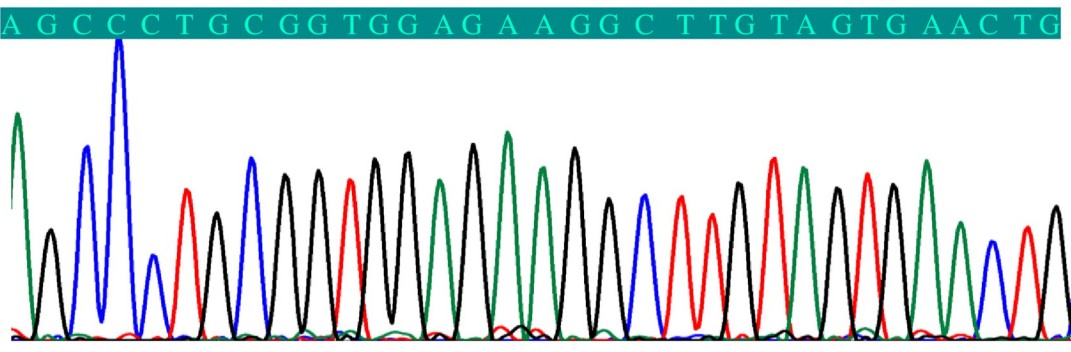

**B**

H-Y antigen from *Hyla eximia*

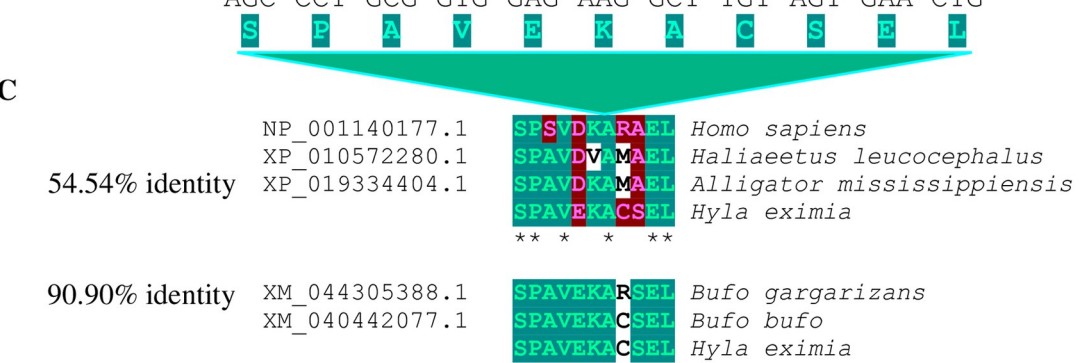

**C**

54.54% identity

| NP_001140177.1 | SPSVDKARAEL | *Homo sapiens* |
| XP_010572280.1 | SPAVDVAMAEL | *Haliaeetus leucocephalus* |
| XP_019334404.1 | SPAVDKAMAEL | *Alligator mississippiensis* |
| | SPAVEKACSEL | *Hyla eximia* |

90.90% identity

| XM_044305388.1 | SPAVEKARSEL | *Bufo gargarizans* |
| XM_040442077.1 | SPAVEKACSEL | *Bufo bufo* |
| | SPAVEKACSEL | *Hyla eximia* |

**D**

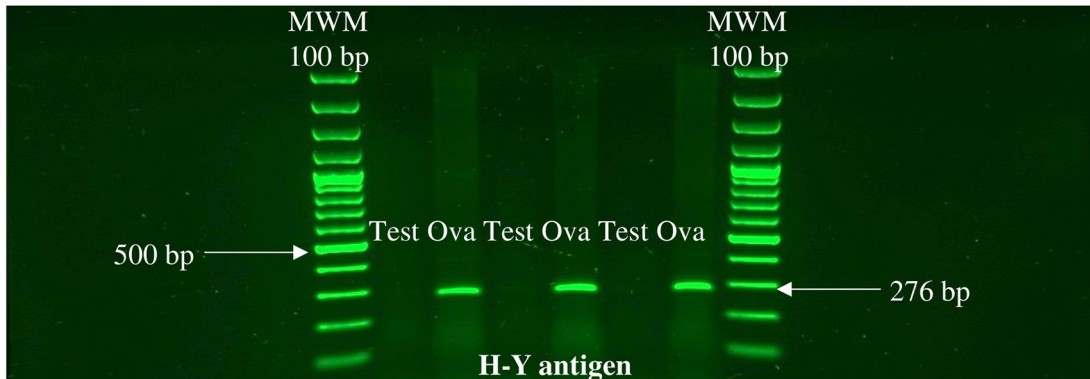

**Fig 2. H-Y antigen sequencing, molecular comparative analysis, and gene expression from *H. eximia* treefrog.** Nucleotide (**A**) and predicted amino acid (**B**) sequence of *H. eximia* H-Y antigen cDNA. The oligonucleotide sequences to amplify the specific cDNA are shown in blue and the H-Y antigen sequence is shown green. (**C**) Comparative analysis of vertebrata H-Y antigen amino acid sequences using ClustalW multiple sequence alignment. The asterisks (*) indicate the same amino acid between sequences. Amino acid residues with a high degree of identity are shown in green, and those with a low degree are shown in red and white. The NCBI database accession numbers of these H-Y antigens are shown to the left of the amino acid sequence. The expression of H-Y antigen inferred the identification of

heterogametic sex using a reverse-transcriptase polymerase chain reaction (RT-PCR) from males (Test = Testis) and female (Ova = Ovary) gonadal tissue during the breeding season. PCR products were compared to a 100 bp MWM (100–1500 bp). Arrows denotes 500 bp and the expected 276 bp PCR product that includes the H-Y antigen (**D**).

```
                                                              ggggtcattcag
       ccggcggcagcggacctgacagcaaagcctattccagcttcttatcgttaaggaaaaccacccccaactcttaagtaaa

ATG GCC GAT GAG GAG ATT GCC GCA CTT GTT GTT GAC AAT GGC TCT GGA ATG TGC AAA GCC    60
 M   A   D   E   E   I   A   A   L   V   V   D   N   G   S   G   M   C   K   A      20
GGC TTT GCT GGA GAT GAT GCT CCC CGT GCT GTC TTC CCC TCC ATT GTG GGT CGC CCA AGA   120
 G   F   A   G   D   D   A   P   R   A   V   F   P   S   I   V   G   R   P   R      40
CAT CAG GGT GTC ATG GTT GGA ATG GGA CAG AAG GAC AGC TAT GTA GGT GAT GAA GCC CAG   180
 H   Q   G   V   M   V   G   M   G   Q   K   D   S   Y   V   G   D   E   A   Q      60
AGC AAG AGG GGT ATC CTG ACC CTG AAG TAC CCC ATT GAA CAC GGA ATT GTC ACC AAC TGG   240
 S   K   R   G   I   L   T   L   K   Y   P   I   E   H   G   I   V   T   N   W      80
GAT GAC ATG GAG AAG ATC TGG CAT CAC ACC TTC TAC AAT GAA CTG CGT GTT GCC CCT GAG   300
 D   D   M   E   K   I   W   H   H   T   F   Y   N   E   L   R   V   A   P   E     100
GAG CAC CCA GTC CTG TTG ACT GAA GCG CCT CTT AAC CCC AAA GCA AAC AGA GAG AAA ATG   360
 E   H   P   V   L   L   T   E   A   P   L   N   P   K   A   N   R   E   K   M     120
ACA CAG ATT ATG TTT GAA ACA TTC AAT ACT CCT GCA ATG TAT GTT GCC ATT CAA GCT GTT   420
 T   Q   I   M   F   E   T   F   N   T   P   A   M   Y   V   A   I   Q   A   V     140
CTG TCC CTG TAT GCT TCT GGA CGT ACA ACT GGT ATT GTG ATG GAC TCT GGT GAT GGT GTC   480
 L   S   L   Y   A   S   G   R   T   T   G   I   V   M   D   S   G   D   G   V     160
ACT CAC ACT GTA CCA ATT TAC GAA GGC TAT GCA CTG CCC CAT GCC ATC CTA CGT CTT GAT   540
 T   H   T   V   P   I   Y   E   G   Y   A   L   P   H   A   I   L   R   L   D     180
TTG GCT GGA CGT GAC CTG ACA GAC TAC CTC ATG AAA ATT CTT ACT GAG AGA GGT TAT AGC   600
 L   A   G   R   D   L   T   D   Y   L   M   K   I   L   T   E   R   G   Y   S     200
TTT ACT ACC ACT GCA GAG AGG GAA ATT GTA CGT GAT ATT AAA GAG AAG CTG TGT TAT GTG   660
 F   T   T   T   A   E   R   E   I   V   R   D   I   K   E   K   L   C   Y   V     220
GCT CTG GAT TTT GAA CAA GAA ATG GCC ACA GCT GCT TCT TCC TCA TCC CTG GAA AAA AGC   720
 A   L   D   F   E   Q   E   M   A   T   A   A   S   S   S   S   L   E   K   S     240
TAT GAG TTG CCT GAT GGG CAG GTT ATA ACT ATT GGC AAT GAG AGG TTC AGA TGT CCA GAA   780
 Y   E   L   P   D   G   Q   V   I   T   I   G   N   E   R   F   R   C   P   E     260
GCA CTC TTC CAG CCT TCC TTC CTT GGA ATG GAA TCC GCC GGT ATC CAT GAA ACC ACT TAC   840
 A   L   F   Q   P   S   F   L   G   M   E   S   A   G   I   H   E   T   T   Y     280
AAC AGC ATC ATG AAA TGT GAC ATT GAT ATC AGG AAG GAT CTG TAT GCC AAC AAT GTA CTG   900
 N   S   I   M   K   C   D   I   D   I   R   K   D   L   Y   A   N   N   V   L     300
TCA GGA GGT ACC ACC ATG TAC CCT GGC ATT GCT GAC AGA ATG CAA AAA GAA ATC ACT GCT   960
 S   G   G   T   T   M   Y   P   G   I   A   D   R   M   Q   K   E   I   T   A     320
TTG GCT CCC AGC ACA ATG AAA ATT AAG ATC ATA GCT CCC CCA GAA CGC AAG TAC TCT GTC  1020
 L   A   P   S   T   M   K   I   K   I   I   A   P   P   E   R   K   Y   S   V     340
TGG ATT GGT GGA TCC ATC CTG GCT TCC CTG TCT ACC TTC CAG CAG ATG TGG ATC AGC AAA  1080
 W   I   G   G   S   I   L   A   S   L   S   T   F   Q   Q   M   W   I   S   K     360
CCA GAA TAC GAT GAA GCT GGC CCC TCC ATT GTG CAC CGC AAA TGC TTT TAA             1128
 P   E   Y   D   E   A   G   P   S   I   V   H   R   K   C   F  ***             376
```

```
atgttcttttcttttatattgtcttcaattccaagatatatcttcaagtgtacaccaagtacatggccaccatttcatt
ccaaaatatgtcttaggctctaattcgataatgtattttttgtttggtttggttggttaggaataaagcttattctattc
ttctattctaaaaaaaaaaaaaaaaaaaaaaaaaaa
```

**Fig 3. Full-length nucleotide and deduced amino acid sequences of *H. eximia* treefrog Actb (GenBank accession number PP107949).** The numbers at the left side of the sequence indicate the position of nucleotide and amino acid of the Actb. The coding region starts with a start codon (ATG) and ends with a stop codon (TAA) is indicated with ***. The putative polyadenylation signal (AATAAA) is underlined. ExPASy translate tool (https://web.expasy.org/translate/) was used to translate nucleotide sequence into amino acid sequence and ExPASy ProtParam tool characterized the molecular/physiochemical parameters of the Actb protein.

in its nucleotide sequence at 139 bp downstream of the stop codon TAA. The open reading frame (ORF) of the Actb cDNA sequence was 1128 bp with 376 deduced amino acids (Fig 3) and contained a calculated molecular weight of 41799.82 with a theoretical isoelectric point of 5.30. The multiple sequence alignment (Fig 4A) revealed that 97% (Fig 4B) of the deduced Actb amino acid sequence were conserved when aligned with Actb of *Homo sapiens* (NP_001092.1), *Haliaeetus leucocephalus* (XP_010568622.1), *Alligator mississippiensis* (XP_014466124.2), *Bufo gargarizans* (XP_044158737.1), and *Rana temporaria* (XP_040213048.1). The three-dimensional (3D) topology of *H. eximia* Actb showed a highly conserved secondary structure compared to other proteins. Similarity analyses identified 21 $\alpha$-helices, 17 $\beta$-strands, and 3 $\gamma$-turns, common structural motifs of ACTBs (Fig 4C).

## Assessment and profiles of H-Y antigen expression

From several treefrog tissue samples, qPCR assays were used to assess the H-Y antigen mRNA expression in males and females during non-breeding and breeding seasons (Fig 5). The heterogametic sex was confirmed to be female based on the gene expression of H-Y antigen from ovaries, and it was detected in high levels (one-way ANOVA test with a *p-value* $< 0.0001$ and a fold-change of 0.2) during non-breeding season; while in male gonad and male/female non-gonadal tissues the mRNA expression was barely detectable. However, the H-Y antigen showed higher expression in female anterior muscles with statistically ($p < 0.05$) significant difference compared to other tissues (Fig 5A). In ovaries, the expression of H-Y antigen transcript was downregulated during breeding season compared to non-breeding season ovary (statistically significant difference = $p < 0.0001$). Nonetheless, the H-Y antigen transcript indicated a significantly higher expression in the female heterogametic sex 24,ZW ($p < 0.0001$). All other tissues (such as lung, brain, vocal sac, anterior and posterior muscles) maintained minimal or null H-Y antigen expression. Therefore, no significant differences between tissues were observed (Fig 5B).

## Discussion

Although amphibian biodiversity is widely represented by treefrogs or leaf-frogs (1055 species, https://amphibiaweb.org/; accessed March 2, 2023), genetic sex-determining and differentiation mechanisms have rarely been explored in them (https://hsd-project.eu/; accessed September 2, 2023). In this regard, it is important to mention that genetic biodiversity or species variability is fundamental in ecosystems equilibrium. Therefore, this study aimed to provide new knowledge about the sex-determining mechanisms (sex chromosomes and heterogametic sex) in treefrog, *H. eximia*, which is a rarely studied species according to previous reports [30, 31].

Based on the morphometric of each chromosome pair, we found a diploid number of 2n = 24 chromosomes, which can be arranged in 12 pairs. Also, experimental assays hypothesized homomorphic sex chromosomes with a female (ZZ/ZW) heterogametic sex-determining system in the *H. eximia* treefrog. Using cytogenetic analyses, we identified the absence of any heteromorphic sex chromosomes in *H. eximia*. We could hypothesize the presence of homomorphic sex chromosomes based on the male and female karyotype. However, on the basis of the method used, we cannot exclude the occurrence of a multiple sex chromosome system [32]. Previous data demonstrated that a more frequent chromosomal finding in amphibians is the presence of homomorphic sex chromosomes (Bufonidae, Pipidae, Dendrobatidae, Ranidae, and Hylidae) and male heterogamety. A heteromorphic sex chromosome pair was identified in a few amphibian species, such as *Crinia bilingual* from Australia [11, 33]. This homomorphic sex-determining system is thought to represent the early stages of sex

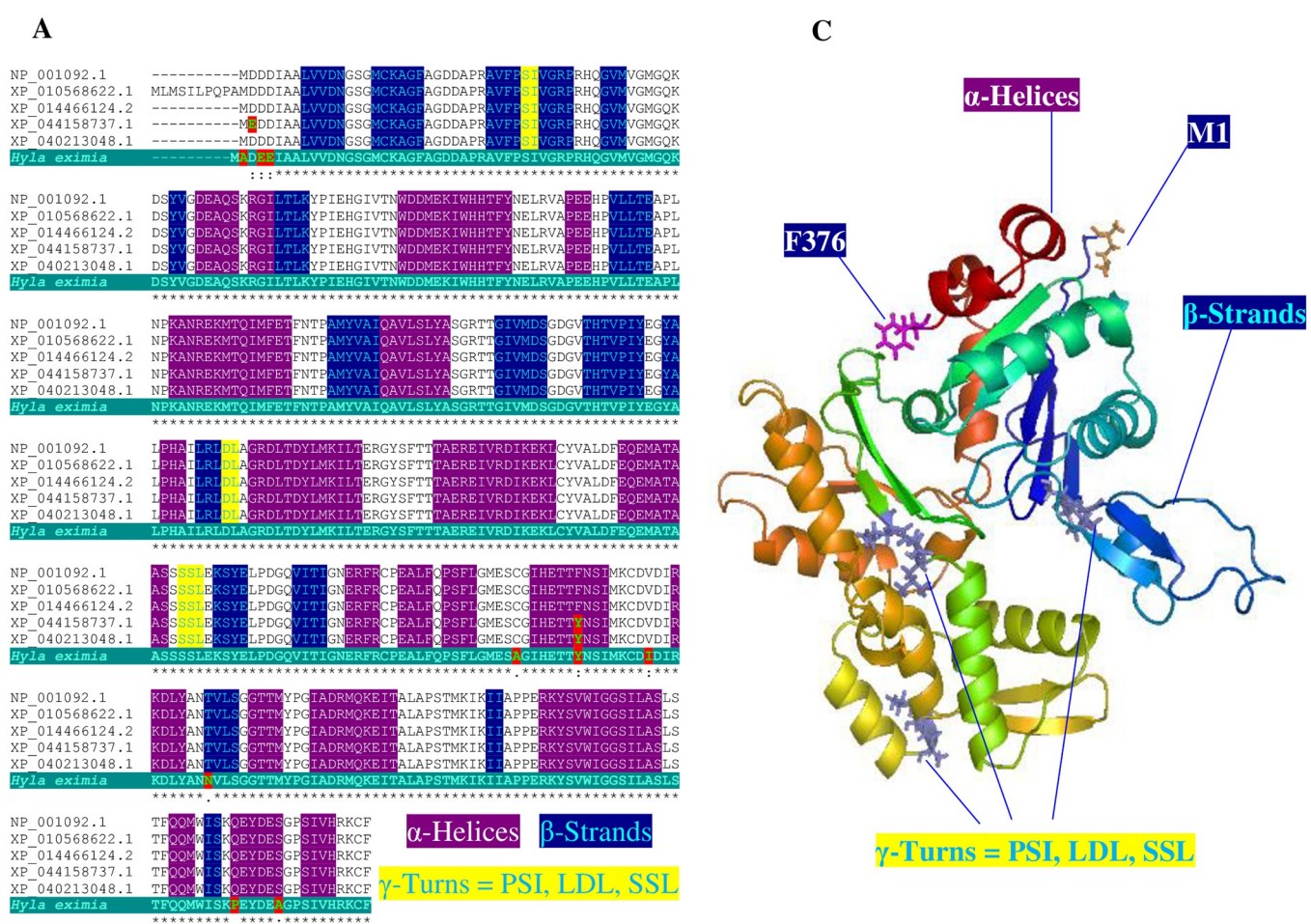

| Species | # Nucleotides | Accession No. | # Amino acids | % identity |
|---|---|---|---|---|
| *Homo sapiens* | 1125 | NP_001092.1 | 375 | 97.6 |
| *Haliaeetus leucocephalus* | 1155 | XP_010568622.1 | 385 | 97.3 |
| *Alligator mississippiensis* | 1125 | XP_014466124.2 | 375 | 97.6 |
| *Bufo gargarizans* | 1125 | XP_044158737.1 | 375 | 97.6 |
| *Rana temporaria* | 1125 | XP_040213048.1 | 375 | 97.9 |
| *Hyla eximia* | 1128 | | 376 | 100 |

**Fig 4. Structural analysis of *H. eximia* treefrog Actb protein.** Multiple sequence alignment of *H. eximia* treefrog Actb (green letters) from deduced amino acid sequences of *Homo sapiens* (NP_001092.1), *Haliaeetus leucocephalus* (XP_010568622.1), *Alligator mississippiensis* (XP_014466124.2), *Bufo gargarizans* (XP_044158737.1), and *Rana temporaria* (XP_040213048.1) was performed using ClustalW software (https://www.genome.jp/tools-bin/clustalw); asterisks indicate similar amino acids between species. Different amino acids are marked with blue/green letters (**A**). Alignment of the predicted *H. eximia* treefrog Actb amino acid sequence with its *B. gargarizans* and *R. temporaria* counterparts showed that these proteins all had different length; the overall homologies of *H. eximia* treefrog Actb with *B. gargarizans* and *R. temporaria* Actb were 97%. Further alignment analyses also showed that *H. eximia* treefrog Actb shared a high percentage of identity with the corresponding Actb of other vertebrates (**B**). The three-dimensional (3D) protein structural prediction analysis was performed via Robetta platform (https://robetta.bakerlab.org/) and PyMol software (https://pymol.org/2/) (**C**).

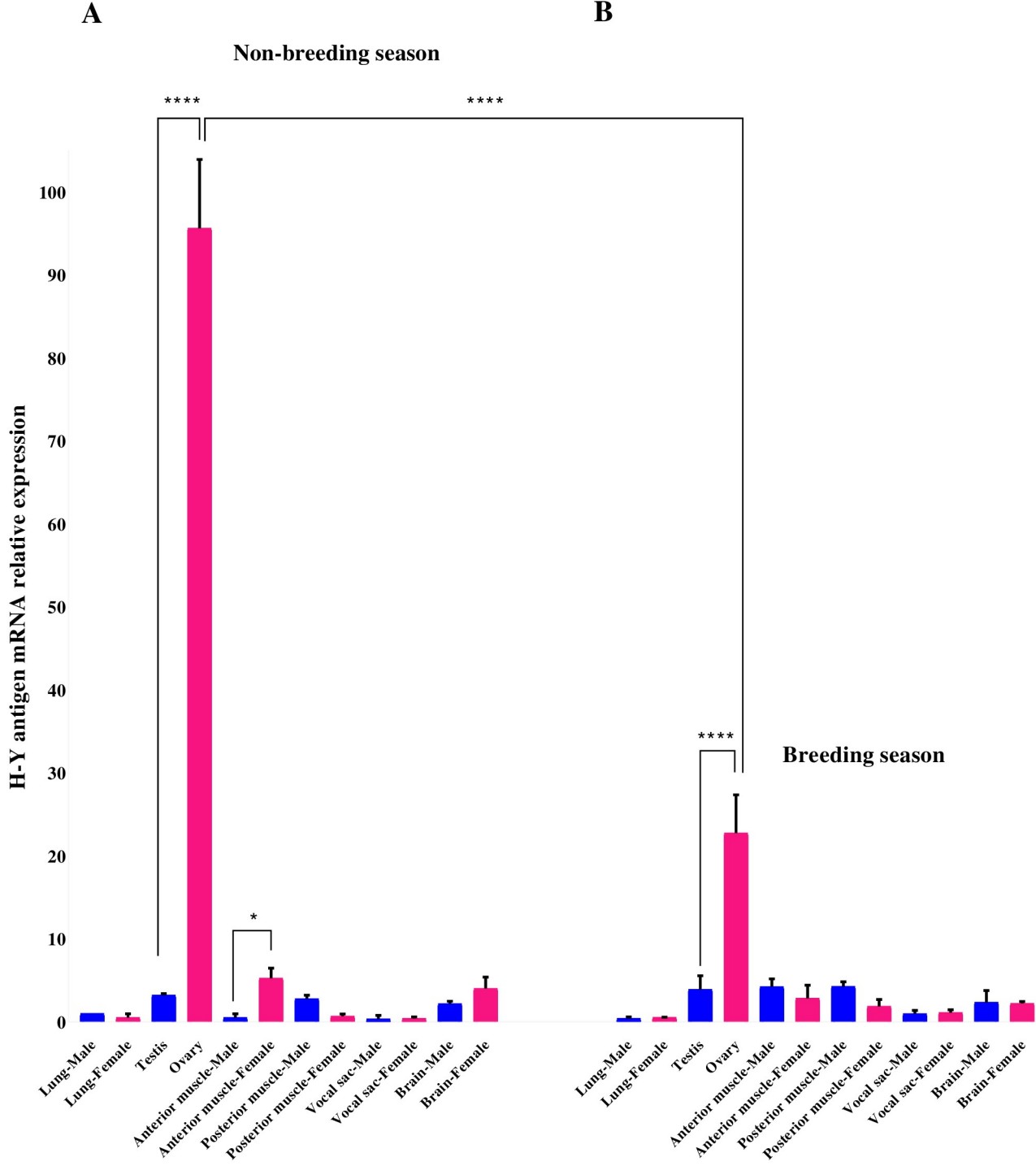

**Fig 5. Involvement of H-Y antigen in the gonad of the heterogametic sex.** Real-time polymerase chain reaction (qPCR) was performed to examine the mRNA levels of H-Y antigen in gonads and several tissues (lung, anterior muscle, posterior muscle, vocal sac, and brain) from adult male and female treefrogs during non-breeding (**A**) and breeding seasons (**B**). Relative gene expression levels were normalized to *β-actin* gene. Vertical bars represent mean values ± standard deviation (SD) of four independent qPCR assays and each bar show male (n = 4; in blue) and female (n = 4; in pink) treefrogs. An analysis of variance (ANOVA) test was used to analyze statistical differences between treefrog groups (non-breeding and breeding seasons; male and female tissues). * and **** indicate $p < 0.05$ and $p < 0.0001$, respectively. $p$-value $< 0.05$ was considered as a significant level statistically.

chromosome evolution. In our research we identified one of the sex-determining factors: potential sex chromosomes and heterogametic sex. We believe that our results could be the initial step toward elucidating and contributing to the genetic sex-determining mechanism in amphibians, specifically in *H. eximia*. Subsequent studies could focus on establishing sex-determining factors or genes.

Also, in this study, the complete cDNA sequences of one reference gene (*Actb*/*β-actin*) were isolated and cloned from *H. eximia*. Actb was cloned and structurally analyzed to carry out the gene expression analyses (qPCR) of H-Y antigen. qPCR studies require a housekeeping gene (in this case, *Actb*) to determine expression levels; in the case of the treefrog, the *Actb* gene has not been cloned or sequenced to date. The nucleotide sequence was used to design primers for qPCR after which the RNA transcript was determined and quantified to analyze the gene expressions profiles in multiple tissues.

To focus on the H-Y transcript expression, gonad samples from the non-breeding and breeding seasons were assessed. We identified that under both conditions H-Y was highly expressed in female treefrogs, suggesting that the H-Y antigen indicates the female heterogametic sex. Interestingly, two inquiries have emerged from this study. First, the physiological consequences of having either a high or low H-Y antigen expression in *H. eximia* needs to be examined. The findings of reduced H-Y expression during the breeding phase suggest that species may be avoiding initiating an immune response thus possibly preventing the generation of specific antibodies against the antigen or the activation of immune cells against male discharge during intercourse that might result in a sex-specific rejection. Conversely, up-regulated expression of H-Y antigen during the non-breeding phase could contribute to sustaining steady physiological functions and safeguarding against potential diseases, foreign substances, and/or pathogens. Second, given the biological context, any subsequent studies need to explore the role of sex steroids coupled to their respective nuclear receptors (NR3C4/AR and NR3A1/ESR1) in regulating the expression of H-Y antigen during the mating phase. The current results suggest that further molecular research is critical for broadening our comprehension of the factors influencing genetic sex in amphibians. Unfortunately, to our knowledge there are no reports about H-Y antigen expression in other species in relation to breeding season, and there are only reports about H-Y antigen detection by immunological assays, without any mention of which breeding season was studied [22].

Although the expression of H-Y antigen was also detected in males, the expression is predominant in females. In this sense, the low expression of male antigen was detected by qPCR, which is a highly sensitive technique. However, the identification of H-Y antigen in males was absent by end-point PCR (Fig 2D). Likewise, several studies carried out using immunological techniques have found a low expression of H-Y antigen in males [25] or could be documented by a differential heterochromatinization between the heterogametic sex chromosomes (ZW) [34]. In summary, H-Y expression in males does not affect the conclusion that the H-Y antigen can be used as a marker of sex and for identifying heterogametic sex in species with homomorphic sex chromosomes.

Immunological detection of the H-Y antigen has been used as a former marker of the heterogametic sex in various vertebrate species (such as *Mus musculus* [23], *Gallus domesticus* [35], *Lebistes reticulatus* [36], *Emys orbicularis* [37], and *Bufo bufo* [22]). These findings led us to draw the conclusion that the H-Y antigen is the product of a gene that is highly conserved throughout vertebrate evolution [22]. However, earlier reports in *G. domesticus* have established that the female gonad tests are positive for the H-Y antigen as the ovary undergoes organotypic differentiation on day 6 of embryonic development. Likewise, it has been demonstrated that female chicken embryo produces estrogens around the time of gonadal differentiation [38]. It has been well established that the exogenous administration of estrogens to

male chicken embryos results in gonadal sex reversal and the transient presence of an ovotestis positive for the H-Y antigen. However, these findings specifically suggest that estrogens regulate H-Y antigen expression in ZW females, potentially serving as a pivotal differentiation factor in the ovogenesis of the avian species *G. domesticus* [35].

In amphibians, the study of these mechanisms provides knowledge about reproductive biology, genetic diversity, variability in the genetic sex-determining mechanism, evolutionary history, and their phylogenetic relationships with other groups of animals. Interestingly, studies on sex determination are crucial for conservation strategies in amphibians. Additionally, these studies provide insights into the sex-determining mechanism in amphibians and could help us understand the underlying mechanisms of sex determination in other vertebrates, including humans.

## Conclusion

In light of the findings concerning these genetic and sex-determining mechanisms, our molecular assays provide evidence that the H-Y antigen can be used directly as a marker of genetic sex and for identifying heterogametic sex in species with homomorphic sex chromosomes. Notably, female treefrog H-Y antigen expression was higher than in males during both the non-breeding and breeding seasons. Likewise, the H-Y antigen expression in females during the non-breeding season was higher than in the breeding season, suggesting that sex steroids and their nuclear receptors might regulate H-Y antigen expression. Nevertheless, further molecular and genetic assays will be needed to evaluate these results and address the remaining sex-determining aspects of the treefrogs.

## Supporting information

**S1 Raw images.**
(PDF)

**S1 File.**
(DOCX)

## Acknowledgments

We would like to thank Dr. Jose Sifuentes-Osornio, Dr. Carlos A Aguilar-Salinas, and Dr. Carlos Arturo Hinojosa Becerril for provided their commitment and support with our research team.

## Author Contributions

**Formal analysis:** Guadalupe Cárdenas.

**Investigation:** Luis Ramos.

**Methodology:** Aidet Ruiz.

**Project administration:** Desiderio Velasco.

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
