## [Decision Letter · Decision Letter 0]

6 Feb 2024

PONE-D-24-02112Genetic sex-determining mechanism in Hyla eximia treefrog inferred from H-Y antigenPLOS ONE

Dear Dr. L,

Thank you for submitting your manuscript to PLOS ONE. After careful consideration, we feel that it has merit but does not fully meet PLOS ONE’s publication criteria as it currently stands. Therefore, we invite you to submit a revised version of the manuscript that addresses the points raised during the review process.

**ACADEMIC EDITOR: **Dear author, review each reviewer's question point by point and send it to me within 20 days.

We look forward to receiving your revised manuscript.

Kind regards,

Mozaniel Santana de Oliveira, Ph.D

Academic Editor

PLOS ONE

Journal Requirements:

3. In your Methods section, please provide additional information regarding the permits you obtained for the work. Please ensure you have included the full name of the authority that approved the field site access and, if no permits were required, a brief statement explaining why

4. We note that your Data Availability Statement is currently as follows: "All relevant data are within the manuscript and its Supporting Information files."

5.  We note that Figure 1 in your submission contain map images which may be copyrighted. All PLOS content is published under the Creative Commons Attribution License (CC BY 4.0), which means that the manuscript, images, and Supporting Information files will be freely available online, and any third party is permitted to access, download, copy, distribute, and use these materials in any way, even commercially, with proper attribution. For these reasons, we cannot publish previously copyrighted maps or satellite images created using proprietary data, such as Google software (Google Maps, Street View, and Earth). For more information, see our copyright guidelines: http://journals.plos.org/plosone/s/licenses-and-copyright.

Reviewers' comments:

Reviewer's Responses to Questions

**Comments to the Author**

1. Is the manuscript technically sound, and do the data support the conclusions?

Reviewer #1: Partly

Reviewer #2: Yes

Reviewer #3: Partly

2. Has the statistical analysis been performed appropriately and rigorously? 

Reviewer #1: N/A

Reviewer #2: Yes

Reviewer #3: Yes

3. Have the authors made all data underlying the findings in their manuscript fully available?

Reviewer #1: Yes

Reviewer #2: No

Reviewer #3: Yes

4. Is the manuscript presented in an intelligible fashion and written in standard English?

Reviewer #1: Yes

Reviewer #2: Yes

Reviewer #3: Yes

5. Review Comments to the Author

Reviewer #1: Ruiz and colleagues reported the study of treefrog sex determination by H-Y antigen, presenting some interesting data, while I have some concerns about the experimental design and data interpretation.

Major concerns:

1. I am confused about the cloning and structural analysis of Actb, what is the relationship of these data with H-Y antigen?

2. H-Y antigen is 33 bp, while the amplicon is 276 bp. I would suggest to show the sequence and primer location, in order to facilitate the readers’ understanding.

3. As shown in Figure 5, H-Y antigen was actually also expressed in male, I wonder if the conclusion could be supported or not.

Minor concerns:

1. In Figure 2C, I suggest to remove Bufo gargarizans and Bufo bufo in upper part

2. Line 58, the sentence “which causes ZZ male to female sex reversal” should be revised. In fish, there are more cases that ZW individuals develop into male under high temperature.

3. Grammatic improvement is required. For example, Line 161, qPCR or quantitative q-PCR, should be streamlined throughout the manuscript.

Reviewer #2: Authors present their work on identification and quantification of the HY-antigen gene in Hyla eximia tree frog. It is a solid piece of work and I enjoyed reading it. There are some major points I would like to raise.

- Title can be toned down a little as authors haven't fully uncovered the sex-determination mechanism. They have identified potential sex chromosomes.

- I found discussion section to be inadequate. Some sections belonged in introduction and results must not be repeated. It is not contextualised broadly in terms of why this study matters to the field. Please rewrite discussion. Discussion should contain information that you could not write before the study was conducted.

One pointer is:

- What are the similarities and differences compared to other species with HY expression in the context of breeding season. Be careful that antigen requires antibodies. So it is antibodies that are relevant and discuss implications from that angle too.

- You can discuss Actb gene and how this can be used as a resource.

Please provide all karyotype images as supplementary information.

Other minor points are listed below:

Abstract:

Line 32: “We also obtained Actb…” – state why it is important. Seems disconnected. Perhaps say that sequence can be used as control for other molecular studies.

Line 34: change qPCR to RT-PCR

Introduction:

Line 49: Please remove “or polygenic”. See https://doi.org/10.1016/j.tig.2022.12.002/ for details.

Line 58-60: Argument for GSD is incorrect. Sry gene can be on autosome and it is sufficient to produce XX males. change it to the gene specific nature and indicate that the sex determining gene generally lies on the sex chromosomes. As per sex-chromosome evolution, first the sex-determining factor evolves and then the sex chromosome forms around it.

Line 63: “Ruiz-Garcia …. commented on” - change the framing of the sentence to “there is a lack of understanding [citation].”

Line 65 and 66: “they commented ……” – This is not true. It is an allele at a locus, generally on the sex chromosome, determine GSD. Process is well understood but not easy to study due to technological limitations. Please correct the framing of the sentence.

Line 82: “SMCY/SMCX genes diverged….” – This is specific to eutherian mammals. Best to be specific when divergence time is mentioned.

Methods:

Line 118: How were cardiacal lymphocytes collected?

There were several instances of 1000 xg. Please change “x” to mathematical symbol.

Primer pairs for HY-antigen in expression profiles were different to the isolation of and expression of H-Y antigen section. Please correct if this is a mistake. If this is not a mistake then justify why primer pairs were different.

Results: Line 210: “morpholically undifferentiated” – change undifferentiated to differentiated.

Line 223: “this observation …” – change to “This implies females are heterogametic sex.”

Line 227: “Despite these results,” is not required here.

Line 229: “Also,” is not required. Please remove that word.

Line 248: “Statistically significant difference…) – Please mention the test, p-value and fold-change in the sentence.

Line 249: “male/female tissues” – change to “male/female non-gonadal tissues” for clarity

Figure 5: what is the unit of measurement for the y-axis. Please state.

Reviewer #3: This manuscript deals with sex determination in the treefrog Hyla eximia.

The work is scientifically sound and concerns a topic of broad scientific interest, however the authors should deal with some issues before it can be accepted for publication.

In particular, it does not seem to me that sex chromosomes were identified in the study species. Differential antigen expression appears to support the hypothesis of a homomorphic sex chromosome system, but the sex chromosome themselves should still be identified (e.g. pair number) cytogenetically and or molecularly. This should be clearly specified in the manuscript.

The introduction needs more information on the study species (e.g. geographic range, phylogeny, previous cytogenetic data etc.) and on sex determination systems in general (ESD, TSD with simple and multiple sex chromosome systems, see below).

Some of the methods should also be better described (e.g. calculation of the centromeric index and chromosome staining and classification, see below).

Here below are some section-specific points that should be carefully addressed.

Introduction

Is this the first karyotype description of the species? Otherwise, the authors should probably add some information in the Introduction on what is currently known on the karyotype of the species and add the relevant citations

Lines 56-60: Also please specify that the cited case of ZZ male to female sex reversal concerns Dragon lizards. That can hardly be generalised for reptiles. In order to avoid confusion, it is probably worth to better describe genetic and environmental sex determination as well as known cases of sex reversal in fish, reptile and amphibians

Lines 56-60: Given the research topic it would be very important to mention that multiple sex chromosome systems (X1X2Y and Z1Z2W) are known in mammals (Saunders and Veyrunes 2021 Genes 8, 12, 1770) reptiles (with either oviparous and viviparous clades) (Petraccioli et al. 2019 Cytogenetic and Genome research, 157, 65-76) and amphibians (Miura et al. 2021 Cells. 10, 661).

Line 68: 95% of described species or 95% of the species with a known karyotype? Please, be more specific.

Line 82: I would invert this sentence in something like “they are probably homologous and likely diverged”

Methods

Line 98: change “from” with “of”

Line 101: The authors should probably add some citations on the breeding season of the species (e.g. Hernández-Salinas et al. 2018, PeerJ., 6, e5897).

Line 107: please describe RNA isolation methods or at least add relevant citations.

Line 131: Please add some information on karyotyping, including chromosome staining techniques, number of images collected per studied sample and if chromosomes were classified based on the centromeric index following Levan et al. (1964) Hereditas 1964, 52, 201–220. Then possibly add calculation of the centromeric index.

Line 134: mention of RNA isolation is repeated in different paragraphs. I suggest to mention and describe it only once in the methods.

Results

Lines 211-212. It does not seem to me that sex chromosomes were identified, for example using C-banding, molecular cytogenetics etc. The authors should specify in the manuscript that the occurrence of homomorphic sex chromosomes is a likely hypothesis based on what is known from other species, but sex chromosome were not cytogenetically identified.

Discussion

Line 285: sex determination is a characteristic of species, not antigens.

Fig 1 caption: this caption should be reworked. Please change “similarities” with something like “the same chromosome number”. Furthermore, 2n = 24 is simply the diploid number, it does not indicate a sex determination system.

6. PLOS authors have the option to publish the peer review history of their article (what does this mean?). If published, this will include your full peer review and any attached files.

Reviewer #1: No

Reviewer #2: **Yes: **Hardip Patel, John Curtin School of Medical Research, Australian National University

Reviewer #3: No

---

## [Author Response · Author response to Decision Letter 0]

10 Apr 2024

Dear Paula Katrina A. Maderazo

PLOS ONE

Thank you for your consideration of our manuscript. We have reviewed each of the comments and responded to each recommendation. The changes are mentioned below and are provided in blue letters.

1.- Data availability was modified to “No – some restrictions will apply”.

2.- The map in Figure 1 was removed.

3.- In Methods section, all information regarding the experiments involving animals was included in detail.

1.- In our manuscript PONE-D-24-02112R1, we did not include maps or satellite images, or images created using proprietary data in Figure 1.

2.- The sacrifice method was included in line 102 “by decapitation”.

3. - The images in Figure 1 were not sourced from an online website. We personally collected the frogs at the location described in the manuscript and photographed one individual on site. These images have not been previously copyrighted and no permissions are required for their use.

RESPONSE TO REVIEWERS

Reviewer #1: 

We greatly appreciate the comments from the reviewer. Likewise, we have kindly considered all the comments to the manuscript and have modified the paper in detail based on all the suggestions. Below, we individually discuss each of your comments. To facilitate this discussion, we first retyped your comments in italic font and then presented our responses in blue.

Ruiz and colleagues reported the study of treefrog sex determination by H-Y antigen, presenting some interesting data, while I have some concerns about the experimental design and data interpretation.

Major concerns:

1. I am confused about the cloning and structural analysis of Actb, what is the relationship of these data with H-Y antigen?

Actb was cloned and structurally analyzed to carry out the gene expression analyses (qPCR) of H-Y antigen. qPCR studies require a housekeeping gene (in this case, Actb) to determine expression levels; in the case of the treefrog, the Actb gene has not been cloned or sequenced to date. This comment was added to the discussion.

2. H-Y antigen is 33 bp, while the amplicon is 276 bp. I would suggest to show the sequence and primer location, in order to facilitate the readers’ understanding.

The oligonucleotide sequences to amplify the specific cDNA are shown in blue (Fig. 2a and Figure legends).

3. As shown in Figure 5, H-Y antigen was actually also expressed in male, I wonder if the conclusion could be supported or not.

Although the expression of H-Y antigen was also detected in males, the expression is predominant in females. In this sense, the low expression of male antigen was detected by qPCR, which is a highly sensitive technique. However, the identification of H-Y antigen in males was absent by end-point PCR (Fig. 2D); likewise, several studies carried out using immunological techniques have found a low expression of H-Y antigen in males (reference 26). In summary, H-Y expression in males does not affect the conclusion. This comment was added to the discussion.

Minor concerns:

In Figure 2C, I suggest to remove Bufo gargarizans and Bufo bufo in upper part.

The nucleotide sequence of Bufo gargarizans and Bufo bufo was eliminated.

Line 58, the sentence “which causes ZZ male to female sex reversal” should be revised. In fish, there are more cases that ZW individuals develop into male under high temperature.

The phrase “in a dragon lizard” was added for the purpose of clarifying the text.

Grammatic improvement is required. For example, Line 161, qPCR or quantitative q-PCR, should be streamlined throughout the manuscript.

Grammar was improved throughout the manuscript.

Reviewer #2:

We greatly appreciate your detailed feedback, and hope that our explanation has fully addressed your concerns. We believe that - thanks to the reviewers’ observations – that the manuscript has been greatly improved. Below, we individually discuss each of your comments. To facilitate this discussion, we first retyped your comments in italic font and then presented our responses in blue.

Authors present their work on identification and quantification of the HY-antigen gene in Hyla eximia tree frog. It is a solid piece of work and I enjoyed reading it. There are some major points I would like to raise.

- Title can be toned down a little as authors haven't fully uncovered the sex-determination mechanism. They have identified potential sex chromosomes.

The title was modified according to one reviewer’s comment about “Understanding the genetic sex-determining mechanism in Hyla eximia treefrog inferred from H-Y antigen”. Yes, we totally agree; in our research we identified one of the sex-determining factors: potential sex chromosomes and heterogametic sex. We believe that our results could be the initial step toward elucidating and contributing to the genetic sex-determining mechanism in amphibians, specifically in H. eximia. Subsequent studies could focus on establishing sex-determining factors or genes. This observation was added to the discussion.

I found discussion section to be inadequate. Some sections belonged in introduction and results must not be repeated. It is not contextualized broadly in terms of why this study matters to the field. Please rewrite discussion. Discussion should contain information that you could not write before the study was conducted.

In amphibians, the study of these mechanisms provides knowledge about reproductive biology, genetic diversity, variability in the genetic sex-determining mechanism, evolutionary history, and their phylogenetic relationships with other groups of animals. Interestingly, studies on sex determination are crucial for conservation strategies in amphibians. Additionally, these studies provide insights into the sex-determining mechanism in amphibians and could help us understand the underlying mechanisms of sex determination in other vertebrates, including humans. This paragraph was added to the discussion.

One pointer is:

- What are the similarities and differences compared to other species with HY expression in the context of breeding season? Be careful that antigen requires antibodies. So it is antibodies that are relevant and discuss implications from that angle too.

Unfortunately, to our knowledge there are no reports about H-Y antigen expression in other species in relation to breeding season, and there are only reports about H-Y antigen detection by immunological assays, without any mention of which breeding season was studied [22]. This paragraph was added to the discussion.

- You can discuss Actb gene and how this can be used as a resource.

The comment was added to the discussion.

Please provide all karyotype images as supplementary information.

All karyotype images were provided as supplementary information.

Other minor points are listed below:

Abstract:

Line 32: “We also obtained Actb…” – state why it is important. Seems disconnected. Perhaps say that sequence can be used as control for other molecular studies.

The sentence “To obtain gene expression profiles, we also obtained the coding sequence of the housekeeping Actb gene” was added.

Line 34: change qPCR to RT-PCR

Line 34 was rewritten.

Introduction:

Line 49: Please remove “or polygenic”. See https://doi.org/10.1016/j.tig.2022.12.002/ for details.

The phrase “or polygenic” was removed.

Line 58-60: Argument for GSD is incorrect. Sry gene can be on autosome and it is sufficient to produce XX males. Change it to the gene specific nature and indicate that the sex determining gene generally lies on the sex chromosomes. As per sex-chromosome evolution, first the sex-determining factor evolves and then the sex chromosome forms around it.

The argument was eliminated.

Line 63: “Ruiz-Garcia …. commented on” - change the framing of the sentence to “there is a lack of understanding [citation].”

The phrase was modified to say, “Ruiz-Garcia et al. [11] commented that there is a lack of understanding about the sex chromosomes”.

Line 65 and 66: “they commented ……” – This is not true. It is an allele at a locus, generally on the sex chromosome, determine GSD. Process is well understood but not easy to study due to technological limitations. Please correct the framing of the sentence.

The sentence was eliminated.

Line 82: “SMCY/SMCX genes diverged….” – This is specific to eutherian mammals. Best to be specific when divergence time is mentioned.

The sentence was modified to say, “In eutherian mammals, both SMCY/SMCX genes diverged at least 120 million years ago and are highly homologous to each other”.

Methods:

Line 118: How were cardiacal lymphocytes collected?

By cardiac puncture, karyotyping was performed using 30 µL of lymphocytes collected in sodium heparin.

There were several instances of 1000 xg. Please change “x” to mathematical symbol.

The mathematical symbol was included.

Primer pairs for HY-antigen in expression profiles were different to the isolation of and expression of H-Y antigen section. Please correct if this is a mistake. If this is not a mistake then justify why primer pairs were different.

This is not a mistake; all primer pairs were different. The oligonucleotides used in qPCR are specifically designed for accurate detection and quantification of gene dosage or gene expression in real time, which requires special design considerations and optimization compared to conventional PCR. The aim is to minimize the formation of primer dimers while creating the optimal concentration of primers and probes. These probes are designed to hybridize specifically to the amplified fragment (~100 bp) during qPCR and release a fluorescent signal. This process allows real-time detection of the progress of gene amplification.

Results: Line 210: “morphologically undifferentiated” – change undifferentiated to differentiated.

The phrase was modified.

Line 223: “this observation …” – change to “This implies females are heterogametic sex.”

The phrase was changed.

Line 227: “Despite these results,” is not required here.

The phrase was eliminated.

Line 229: “Also,” is not required. Please remove that word.

The word was removed.

Line 248: “Statistically significant difference…) – Please mention the test, p-value and fold-change in the sentence.

The sentence has been changed to say “one-way ANOVA test with a p-value < 0.0001 and a fold-change of 0.2”. 

Line 249: “male/female tissues” – change to “male/female non-gonadal tissues” for clarity.

The phrase has been changed.

Figure 5: what is the unit of measurement for the y-axis. Please state.

The Y-axis does not have units of measurement and only represents the mRNA relative expression. Generally, the Y-axis represents the fluorescence signal measured during the amplification process. This signal comes from a fluorescence probe that specifically binds to the PCR product of the gene of interest and the housekeeping gene as it is amplified during the reaction cycle. Therefore, the Y-axis represents the ratio of fluorescence intensity measured in each PCR cycle.

Reviewer #3:

We greatly appreciate all of the comments from the reviewer, and we believe that - thanks to the reviewer’s observations - the manuscript has been much improved. Below, we individually discuss each of your comments. To facilitate this discussion, we first retyped your comments in italic font and then presented our responses in blue.

This manuscript deals with sex determination in the treefrog Hyla eximia.

The work is scientifically sound and concerns a topic of broad scientific interest, however the authors should deal with some issues before it can be accepted for publication.

In particular, it does not seem to me that sex chromosomes were identified in the study species. Differential antigen expression appears to support the hypothesis of a homomorphic sex chromosome system, but the sex chromosome themselves should still be identified (e.g. pair number) cytogenetically and or molecularly. This should be clearly specified in the manuscript.

Introduction

Is this the first karyotype description of the species? Otherwise, the authors should probably add some information in the Introduction on what is currently known on the karyotype of the species and add the relevant citations.

To our knowledge, this is the first karyotype description of H. eximia.

Lines 56-60: Also please specify that the cited case of ZZ male to female sex reversal concerns Dragon lizards. That can hardly be generalised for reptiles. In order to avoid confusion, it is probably worth to better describe genetic and environmental sex determination as well as known cases of sex reversal in fish, reptile and amphibians.

The phrase “in a dragon lizard” was included for the purpose of clarifying the text.

Lines 56-60: Given the research topic it would be very important to mention that multiple sex chromosome systems (X1X2Y and Z1Z2W) are known in mammals (Saunders and Veyrunes 2021 Genes 8, 12, 1770) reptiles (with either oviparous and viviparous clades) (Petraccioli et al. 2019 Cytogenetic and Genome research, 157, 65-76) and amphibians (Miura et al. 2021 Cells. 10, 661).

This paragraph was included in the introduction: “In addition, multiple sex chromosome systems (X1X2Y and Z1Z2W) have been reported in mammals [8], reptiles (with both oviparous and viviparous clades) [9], and amphibians [10]. These data highlight the variability in chromosomic and genetic sex-determining mechanisms in vertebrate groups”.

Line 68: 95% of described species or 95% of the species with a known karyotype? Please, be more specific.

The sentence was changed.

Line 82: I would invert this sentence in something like “they are probably homologous and likely diverged”

The sentence was changed.

Methods

Line 98: change “from” with “of”

The word was changed.

Line 101: The authors should probably add some citations on the breeding season of the species (e.g. Hernández-Salinas et al. 2018, PeerJ., 6, e5897).

The reference was added.

Line 107: please describe RNA isolation methods or at least add relevant citations.

The RNA isolation method is now mentioned.

Line 131: Please add some information on karyotyping, including chromosome staining techniques, number of images collected per studied sample and if chromosomes were classified based on the centromeric index following Levan et al. (1964) Hereditas 1964, 52, 201–220. Then possibly add calculation of the centromeric index.

Information on karyotyping was described as follows: “The chromosomes were stained with Giemsa (1:10) for 20 min. The chromosome analysis was performed using 15 metaphases for each sex (n = 15 for males and n = 15 for females) in the 100X immersion objective. Karyotyping was analyzed by the automated cell imaging system Leica Biosystems, using Cytovision DM2500. The chromosomes were analyzed with ImageJ software and classified based on the centromeric index (CI) according to Levan et al. [27]”.

Line 134: mention of RNA isolation is repeated in different paragraphs. I suggest to mention and describe it only once in the methods.

RNA isolation is now mentioned and described in the methods.

Results

Lines 211-212. It does not seem to me that sex chromosomes were identified, for example using C-banding, molecular cytogenetics etc. The authors should specify in the manuscript that the occurrence of homomorphic sex chromosomes is a likely hypothesis based on what is known from other species, but sex chromosome were not cytogenetically identified.

We did not identify sex chromosomes. No heteromorphic sex chromosomes (morphologically differentiated) were identified based on either size or shape in the two sexes of H. eximia. We could hypothesize the presence of homomorphic sex chromosomes based on the male and female karyotype. This last sentence was added to the discussion.

Discussion

Line 285: sex determination is a characteristic of species, not antigens.

The phrase was modified.

Fig 1 caption: this caption should be reworked. Please change “similarities” with something like “the same chromosome number”. Furthermore, 2n = 24 is simply the diploid number, it does not indicate a sex determination system.

Figure 1 was modified, and we elimin

---

## [Decision Letter · Decision Letter 1]

26 Apr 2024

PONE-D-24-02112R1Understanding the genetic sex-determining mechanism in Hyla eximia treefrog inferred from H-Y antigenPLOS ONE

Dear Dr. Ramos L,

Thank you for submitting your manuscript to PLOS ONE. After careful consideration, we feel that it has merit but does not fully meet PLOS ONE’s publication criteria as it currently stands. Therefore, we invite you to submit a revised version of the manuscript that addresses the points raised during the review process.

**ACADEMIC EDITOR: **Dear authors, I have received the reminders of the reviews, please review each question point by point.. 

We look forward to receiving your revised manuscript.

Kind regards,

Mozaniel Santana de Oliveira, Ph.D

Academic Editor

PLOS ONE

Journal Requirements:

Reviewers' comments:

Reviewer's Responses to Questions

**Comments to the Author**

1. If the authors have adequately addressed your comments raised in a previous round of review and you feel that this manuscript is now acceptable for publication, you may indicate that here to bypass the “Comments to the Author” section, enter your conflict of interest statement in the “Confidential to Editor” section, and submit your "Accept" recommendation.

Reviewer #2: All comments have been addressed

Reviewer #3: (No Response)

2. Is the manuscript technically sound, and do the data support the conclusions?

Reviewer #2: Yes

Reviewer #3: Partly

3. Has the statistical analysis been performed appropriately and rigorously? 

Reviewer #2: Yes

Reviewer #3: (No Response)

4. Have the authors made all data underlying the findings in their manuscript fully available?

Reviewer #2: Yes

Reviewer #3: Yes

5. Is the manuscript presented in an intelligible fashion and written in standard English?

Reviewer #2: No

Reviewer #3: Yes

6. Review Comments to the Author

Reviewer #2: There are many cases where semicolon is used. These can generally be replaced with a full-stop. Copy editing of this manuscript is essential to ensure readability. I will not go through this one-by-one as it is too onerous. Authors and journals should be responsible for the correct use of the language.

Reviewer #3: In my opinion, the authors have done a good job in revising several parts of the manuscript which now are more accurate and sufficiently detailed.

However, a further revision of some limited but important points should be performed before the manuscript can be accepted for publication.

Line 203-204: The results obtained point at a probable genetic sex determination system with female heterogamety. However, a chromosome complement with either ZW in females or ZZ in male was not identified. In fact, on the basis of the method used, the author cannot exclude the occurrence of a multiple sex chromosome system (see e.g Gazoni et al 2018. Chromosoma 127(2):269-278).

Line 207: Change “karyotyping” with “karyotype”

Line 209: Change “karyotypic morphology” with “chromosome morphology”

Line 209: insert “pairs” before “1” and “5” and “4”. Remove “chromosome” in lines 211 and 212.

Line 228: change with something like “represent the heterogametic sex”.

Line 263: add something like “in them” after “explored”

Line 269-273: I think there is still some confusion here. In fact, the presence sex chromosome system can be either “revealed” (as suggested at line 270) or “hypothesized” (as stated at line 272). The results obtained with the methods employed by the author allow them to just hypothesize the presence of a genetic ZW sex chromosome system. Please, be more precise with similar statements in several sections of the manuscript in order to be more consistent with the results obtained and avoid confusion for the general reader.

Line 312-314: I think that it would be important to highlight that the hypothesized occurrence of sex chromosome system could be documented by a differential heterochromatinization between the heterogametic sex chromosomes (ZW) (Mezzasalma et al. 2007 Salamandra 55,:140-144).

7. PLOS authors have the option to publish the peer review history of their article (what does this mean?). If published, this will include your full peer review and any attached files.

Reviewer #2: **Yes: **Hardip R. Patel

Reviewer #3: No

---

## [Author Response · Author response to Decision Letter 1]

29 Apr 2024

Response to Reviewer #2 and Reviewer #3 (April 29-2024):

We greatly appreciate the comments from the reviewers. Likewise, we have kindly considered all the comments to the manuscript and have modified the paper in detail based on all suggestions. We believe the manuscript is now much improved. All changes to the revised paper are indicated by pink text.

---

## [Decision Letter · Decision Letter 2]

15 May 2024

Understanding the genetic sex-determining mechanism in Hyla eximia treefrog inferred from H-Y antigen

PONE-D-24-02112R2

Dear Dr. Ramos,

We’re pleased to inform you that your manuscript has been judged scientifically suitable for publication and will be formally accepted for publication once it meets all outstanding technical requirements.

Kind regards,

Mozaniel Santana de Oliveira, Ph.D

Academic Editor

PLOS ONE

Additional Editor Comments (optional):

Reviewers' comments:

Reviewer's Responses to Questions

**Comments to the Author**

1. If the authors have adequately addressed your comments raised in a previous round of review and you feel that this manuscript is now acceptable for publication, you may indicate that here to bypass the “Comments to the Author” section, enter your conflict of interest statement in the “Confidential to Editor” section, and submit your "Accept" recommendation.

Reviewer #3: All comments have been addressed

2. Is the manuscript technically sound, and do the data support the conclusions?

Reviewer #3: Yes

3. Has the statistical analysis been performed appropriately and rigorously? 

Reviewer #3: Yes

4. Have the authors made all data underlying the findings in their manuscript fully available?

Reviewer #3: Yes

5. Is the manuscript presented in an intelligible fashion and written in standard English?

Reviewer #3: Yes

6. Review Comments to the Author

Reviewer #3: The authors have submitted a revised version of the manuscript which feels improved and more accurate.

I have no further comments.

7. PLOS authors have the option to publish the peer review history of their article (what does this mean?). If published, this will include your full peer review and any attached files.

Reviewer #3: No

---

## [Editor Report · Acceptance letter]

17 May 2024

PONE-D-24-02112R2 

PLOS ONE

Dear Dr. Ramos, 

I'm pleased to inform you that your manuscript has been deemed suitable for publication in PLOS ONE. Congratulations! Your manuscript is now being handed over to our production team.

Kind regards, 

on behalf of

Dr. Mozaniel Santana de Oliveira 

Academic Editor

PLOS ONE